# A Phase Ib Study of Durvalumab (MEDI4736) in Combination with Carbon-Ion Radiotherapy and Weekly Cisplatin for Patients with Locally Advanced Cervical Cancer (DECISION Study): The Early Safety and Efficacy Results

**DOI:** 10.3390/ijms241310565

**Published:** 2023-06-23

**Authors:** Noriyuki Okonogi, Kazutoshi Murata, Shigeru Yamada, Yuji Habu, Makoto Hori, Tomoya Kurokawa, Yosuke Inaba, Tadami Fujiwara, Yasuhisa Fujii, Michiko Hanawa, Yohei Kawasaki, Yoko Hattori, Kazuko Suzuki, Kyoko Tsuyuki, Masaru Wakatsuki, Masashi Koto, Sumitaka Hasegawa, Hitoshi Ishikawa, Hideki Hanaoka, Makio Shozu, Hiroshi Tsuji, Hirokazu Usui

**Affiliations:** 1QST Hospital, National Institutes for Quantum Science and Technology, 4-9-1 Anagawa, Inage-ku, Chiba 263-8555, Japan; okonogi.noriyuki@qst.go.jp (N.O.); murata.kazutoshi@qst.go.jp (K.M.); suzuki.kazuko@qst.go.jp (K.S.); tsuyuki.kyoko@qst.go.jp (K.T.); wakkun100@gmail.com (M.W.); koto.masashi@qst.go.jp (M.K.); ishikawa.hitoshi@qst.go.jp (H.I.); tsuji.hiroshi@qst.go.jp (H.T.); 2Department of Reproductive Medicine, Chiba University Graduate School of Medicine, 1-8-1 Inohana, Chuo-ku, Chiba 260-8670, Japan; habuyujp@yahoo.co.jp (Y.H.); shozumakio@chiba-u.jp (M.S.); hirokazu-usui@faculty.chiba-u.jp (H.U.); 3Clinical Research Center, Chiba University Hospital, 1-8-1 Inohana, Chuo-ku, Chiba 260-8677, Japan; makotohori@chiba-u.jp (M.H.); t-kurokawa@chiba-u.jp (T.K.); y.inaba@chiba-u.jp (Y.I.); t-fujiwara@chiba-u.jp (T.F.); fujiiya1@faculty.chiba-u.jp (Y.F.); m-hanawa@chiba-u.jp (M.H.); hattori@chiba-u.jp (Y.H.); hanaoka.hideki@faculty.chiba-u.jp (H.H.); 4Faculty of Nursing, Japanese Red Cross College of Nursing, 4-1-3 Hiroo, Shibuya-Ku, Tokyo 150-0012, Japan; y-kawasaki@redcross.ac.jp; 5Department of Charged Particle Therapy Research, National Institutes for Quantum Science and Technology, 4-9-1 Anagawa, Inage-ku, Chiba 263-8555, Japan; hasegawa.sumitaka@qst.go.jp

**Keywords:** clinical trial, radiotherapy, heavy ion radiotherapy, cisplatin, durvalumab, uterine cervical neoplasms, carbon-ion radiotherapy, anti-PD-L1 antibody, concurrent chemoradiotherapy

## Abstract

We conducted a phase Ib study to examine the safety of a combination of carbon-ion RT (CIRT) with durvalumab (MEDI4736; AstraZeneca) in patients with locally advanced cervical cancer. This was an open-label, single-arm study with a modified 3 + 3 design. Patients with newly diagnosed histologically proven locally advanced cervical cancer were enrolled. All patients received 74.4 Gy of CIRT in 20 fractions and concurrent weekly cisplatin (chemo-CIRT) at a dose of 40 mg/m^2^. Durvalumab was administered (1500 mg/body) at weeks two and six. The primary endpoint was the incidence of adverse events (AEs) and serious AEs (SAEs), including dose-limiting toxicity (DLT). All three enrolled patients completed the treatment without interruption. One patient developed hypothyroidism after treatment and was determined to be an SAE. No other SAEs were observed. The patient recovered after levothyroxine sodium hydrate treatment. None of the AEs, including hypothyroidism, were associated with DLT in the present study. All three patients achieved complete responses within the CIRT region concerning treatment efficacy. This phase 1b trial demonstrates the safety of combining chemo-CIRT and durvalumab for locally advanced cervical cancer in the early phase. Further research is required as only three patients were included in this study.

## 1. Introduction

Cervical cancer is the fourth most common cancer in women worldwide. In 2020, there were approximately 604,000 new cases and 342,000 deaths [1]. For locally advanced cervical cancer, concurrent chemoradiotherapy (CCRT) is the standard treatment strategy [2,3,4,5]. Intracavitary brachytherapy is a critical component of radiotherapy (RT) in patients with cervical cancer [2]. Three-dimensional image-guided brachytherapy (3D-IGBT) has become a critical strategy, and a recent prospective clinical trial demonstrated a favorable local control (LC) rate [6,7,8]. However, recent reports have indicated that patients with bulky tumors or adenocarcinomas have a lower LC rate even when 3D-IGBT is employed [9,10,11]. Therefore, these types of cervical cancer require novel therapeutic approaches.

Charged particle therapies such as proton beam therapy and carbon-ion RT (CIRT) are promising forms of RT with excellent dose distributions. Numerous clinical trials have been conducted using CIRT for many types of tumors and have shown favorable outcomes, taking advantage of the superior dose distribution and biological benefits of higher linear energy transfer (LET) [12]. We have investigated the significance of CIRT for cervical cancer in previous decades. Even in patients with tumor diameters > 6 cm, we have shown a five-year LC rate of 70% for cervical squamous cell carcinoma [13]. Wakatsuki et al. reported a five-year LC rate of 55% for locally advanced adenocarcinoma of the uterine cervix [14]. Subsequently, concurrently administering weekly cisplatin with CIRT for locally advanced adenocarcinoma of the uterine cervix showed promising clinical outcomes, with two-year LC and overall survival (OS) rates of 71% and 88%, respectively [15]. We also reported the significance of the concurrent use of cisplatin in CIRT for locally advanced adenocarcinoma of the uterine cervix, resulting in improved OS and distant metastasis-free rates compared to CIRT alone, using a propensity score-matched analysis [16]. According to a recent systematic review, CIRT is both safe and effective for cervical cancer [17]. However, owing to the aggressive nature of metastatic potential, distant metastasis is still observed even when concurrent chemo-CIRT is used for locally advanced cervical cancer. Thus, to improve the clinical outcomes of difficult-to-treat uterine cervical cancers, such as bulky tumors or adenocarcinomas, a new strategy to prevent distant metastasis is required.

The immune system recognizes cancer cells and sometimes controls or eliminates them. Based on this concept, immune checkpoint inhibitor (ICI) therapy has ushered in a new era of antitumor treatment, with sustained responses and significant survival advantages observed in various tumors. The coordination of programmed death-ligand 1 (PD-L1) and programmed death-1 (PD-1) receptors play critical roles in the T-cell immune response in several molecules involved in the antitumor immune response [18]. When PD-L1 binds to PD-1, T cells receive an inhibitory signal, which decreases their proliferation and cytokine production. Tumor cells utilize this immune checkpoint pathway to escape immune surveillance [19]. One of the ICIs, an anti-programmed death-ligand 1 (PD-L1) antibody, has been shown to be effective in treating various cancers in clinical settings [20,21,22,23]. The “PACIFIC Trial” was a symbolic success, particularly in combining radiotherapy and an ICI [20,21]. Durvalumab, an anti-PD-L1 antibody used as an adjuvant after CCRT for non-small cell lung cancer (NSCLC), improved OS and progression-free survival [20,21]. Thus, the combination of durvalumab and CCRT appears to be a promising treatment strategy.

Many preclinical and clinical studies have shown that RT activates antitumor immunity [24]. However, we recently discovered that CIRT for patients with cervical cancer upregulated PD-L1 expression in tumor tissue samples [25]. This could imply that the antitumor effect of CIRT was suppressed by PD-L1 upregulation. Therefore, combining anti-PD-L1 antibody and CIRT is expected to maximize the local effect of CIRT while inhibiting distant metastasis through activated antitumor immunity. However, there have been no reports on the combination of CIRT and durvalumab. Therefore, we conducted a phase Ib study to test the safety and efficacy of this combination in patients with locally advanced cervical cancer. This report aims to demonstrate the initial efficacy and safety of CIRT with durvalumab in patients with locally advanced cervical cancer.

## 2. Results

### 2.1. Patient Population

Three patients were included in the study, although the initial plan was to enroll a maximum of ten patients. After enrolling three patients, the dramatic rise in cases of coronavirus disease 2019 (COVID-19) in Japan made further enrollment difficult. Specifically, if a patient or the patient’s family had a fever exceeding 37.5 °C, proof of a negative PCR test for COVID-19 was required at the time of the visit to our hospital. Therefore, we decided to terminate enrollment in the study up to the enrollment of three patients. Table 1 summarizes the clinical characteristics of the three patients.

### 2.2. Treatment Feasibility and Adverse Events

All enrolled patients completed the prescribed course of treatment, which included 74.4 Gy of CIRT, five courses of 40 mg/m^2^ cisplatin weekly, and two courses of 1500 mg/body durvalumab administered at weeks two and six. No treatment delays were observed. All treated patients complied with the guidelines for good clinical practice (GCP) and had no protocol violations; thus, safety and efficacy were assessed in these three patients.

Table 2 summarizes the adverse events (AEs) associated with treatment. All patients had grade 3 neutropenia, and one patient (33%) had increased gamma-glutamyl transpeptidase levels. These toxicities improved rapidly after treatment. One patient developed hypothyroidism after treatment. The patient had an increase in free T3 (5.23 pg/mL: normal level of 2.52–4.06 pg/mL) and a decrease in thyroid-stimulating hormone (TSH) (0.07 µIU/mL: normal level of 0.61–4.23 µIU/mL) two months after starting treatment. Subsequently, a decrease in free T3 (1.97 pg/mL) and free T4 (0.40 ng/dL—normal level of 0.75–1.45 ng/dL), and an increase in TSH (24.3 µIU/mL) were observed 14 weeks after the start of treatment. The patient showed no abnormal physical signs. A further decrease in free T3 (<0.67 pg/mL) and free T4 (<0.10 ng/dL) and a further increase in TSH (≥200 µIU/mL) were observed six months after the start of treatment, along with fatigue and facial edema. Based on the course and symptoms, the patient was diagnosed with hypothyroidism. After hospitalization, the patient recovered after treatment with levothyroxine sodium hydrate. None of the AEs, including hypothyroidism, were associated with dose-limiting toxicity (DLT) in the present study. In other words, none of the three patients showed DLT.

All three patients achieved complete response (CR) within the CIRT region concerning treatment efficacy. Thus, the objective response (OR) and CR rates in this study were 100%. Figure 1 shows representative images of the patients who received treatment. For lesions outside the CIRT irradiation field, one patient developed para-aortic lymph node (PAN) metastasis 42 weeks after the start of treatment. There was no metastasis to other sites, and additional treatment was decided upon for metastatic PAN.

## 3. Discussion

To our knowledge, this is the first clinical trial to combine CIRT and durvalumab. Although many clinical trials on the combination of RT and ICI have been conducted, this is the first clinical trial to combine CIRT and ICI in the world, and the concurrent use of ICI and CIRT is also highly novel. All three patients achieved CR within the CIRT region, and none developed DLT. Although only three patients were included, the combination of chemo-CIRT and durvalumab appears promising, considering its initial efficacy and safety profile.

There is no doubt that ICIs are effective against many types of cancer. In addition, the clinical significance of the combination of ICI and RT is gaining support [20,21,26,27]. To date, in phase III trials that have shown positive results with the combination of ICI and RT, ICI has been performed as an adjuvant therapy to RT [20,21,27]. However, the optimal timing of ICI therapy for RT remains controversial. Sato et al. reported that the DNA double-strand break repair pathway regulates PD-L1 expression in cancer cells [28]. Their report showed the upregulation of PD-L1 on the tumor cell surface 24–48 h after irradiation [28]. We found a significant upregulation of PD-L1 expression one week after starting CIRT in clinical cervical cancer specimens [25]. These results suggest that irradiation enhances PD-L1 expression on the tumor surface within a few days of the start of irradiation. Considering these facts, it would make sense to use ICI, especially anti-PD-L1 antibody, simultaneously with RT, as in this study. Currently, a phase III trial of durvalumab in combination with CCRT for NSCLC is ongoing [29]. The results of this trial provide important insights into whether durvalumab should be used concurrently with CCRT or as an adjuvant.

Although CIRT has been suggested to be more effective than conventional RT in treating cervical cancer, particularly adenocarcinoma [30], the clinical benefit of CIRT over conventional RT when used in conjunction with ICI remains unclear. However, CIRT may more efficiently elicit an antitumor immune response. Critically, radiation-elicited T-cell activation is mediated by the accumulation of cytosolic DNA in irradiated cells, with consequent activation of the cyclic GMP-AMP synthase (cGAS)/stimulator of interferon (IFN) genes (STING) pathway and downstream production of type-I IFN and other pro-inflammatory cytokines [31]. CIRT is known to cause complex double-strand breaks in DNA owing to its high LET [32], which may result in the efficient activation of T cells via the cGAS-STING pathway. Although further validation of the clinical significance is needed, an anti-PD-L1 antibody that maintains T-cell activation and CIRT may be an ideal combination.

Although the study enrolled only three patients, all patients completed the scheduled treatment and did not develop DLT. Thus, by combining chemo-CIRT and anti-PD-L1 antibodies, our treatment strategy was considered tolerable. A recent meta-analysis of toxicity in a regimen combining ICIs and RT found that the incidence of grade 3–4 toxicity was comparable to an ICI-alone regimen [33]. The Grade 3 neutropenia was likely due to chemo-CIRT, and the concurrent use of durvalumab did not exacerbate this AE. Our results support the findings of their study. It is worth mentioning that the concurrent use of durvalumab did not worsen the AEs of CIRT in the irradiated region, specifically intestinal disturbances. Careful monitoring of the presence and extent of late AEs in the three cases treated in this study is needed. One patient developed hypothyroidism after the treatment. The possibility of hypothyroidism after durvalumab administration is widely known, and this is an AE that can occur with other ICIs [34]. However, it should be noted that hypothyroidism can occur even with short-term durvalumab administration. In our treatment schedule, durvalumab was administered twice during CIRT and was not used as adjuvant therapy. The patient manifested hypothyroidism between three and six months of treatment. This means that patients should be closely monitored after the last dose of durvalumab, even if only a small number of doses are administered and completed. The three patients will continue to be closely monitored.

The primary limitation of this study is the small number of patients, only three patients; while the impact of COVID-19 on patient aggregation was discouraging, validation in a larger number of patients may be warranted. A phase II trial with a larger number of patients is desired in the future. Another limitation is the possibility of bias due to favorable performance status, as well as the fact that the study was conducted in a single country. Currently, only a few centers offer CIRT, though this number is steadily increasing. CIRT combined with chemotherapy containing ICI will be available in many centers in the future.

## 4. Materials and Methods

### 4.1. Overview of Study Design

This was a phase 1b, interventional, open-label, single-arm study named the DECISION study (Japan Registry of Clinical Trials: jRCT2031210083). The protocol design was described in a previous report [35]. Japanese or non-Japanese patients who fully understood Japanese were allowed to enroll according to a modified 3 + 3 design, and we assessed the safety of the first three patients. Seven additional patients were planned to be enrolled if no DLT developed in these three patients. If one of the first three patients developed DLT, three additional patients were planned to be enrolled. The study was terminated if two or all of the first three patients developed DLT. The period for evaluating DLTs was set from the start of treatment to 92 days after that. No randomization was intended for patient enrollment. Table 3 describes the DLT used to assess the safety of this study. The study protocol was approved by the Human Research Ethics Committees of QST Hospital (#C21-002, 26 April 2021) and Chiba University (#2021006, 21 April 2021). This study was carried out in accordance with the Helsinki Declaration’s principles.

### 4.2. Protocol Treatment

All patients received 74.4 Gy of CIRT in 20 fractions and concurrent weekly cisplatin at a dose of 40 mg/m^2^. Durvalumab was administered (1500 mg/body) at weeks two and six. CIRT and cisplatin were administered at the QST Hospital in a way approved by the responsible ministry and relevant radiotherapy society [36]. The modified microdosimetric kinetic model (MKM) was applied to calculate the relative biological effectiveness (RBE) of calculation at our institution [37]. The “Gy” in this paper is the RBE-weighted dose based on the modified MKM. Durvalumab was administered at the Chiba University Hospital. The dosage of durvalumab was based on the CALLA trial, which compared concurrent and adjuvant durvalumab with CCRT versus CCRT alone in patients with locally advanced cervical cancer [38]. The enrolled patients were followed up strictly per the scheduled protocol for one year after CIRT initiation (Figure 2). Appendix A summarizes the relevant concomitant care and interventions permitted or prohibited during this study.

### 4.3. Patient Eligibility

This study included patients with locally advanced cervical cancer, as defined by the International Federation of Gynecology and Obstetrics (FIGO) 2018 staging: histologically proven uterine cervical cancer of FIGO stage IIB, IIIA, IIIB, IIIC1, or IVA. This study used physical examination and diagnostic imaging to determine staging rather than surgical diagnosis. Appendix A lists the inclusion and exclusion criteria.

### 4.4. Endpoints

The incidence of AEs and serious AEs, including DLTs, was the primary endpoint. The secondary endpoints were one-year OS, progression-free survival, and the distant metastasis rate. In addition, as secondary endpoints, the OR and CR rates were assessed. Furthermore, tumor tissue and blood were collected before and one week after treatment began. These samples were evaluated as an exploratory goal to assess the relationship between immune response and prognosis.

This paper focuses on the early results for safety and efficacy; thus, it reports the AEs and response rates.

### 4.5. Evaluation Methods

All data analyses were performed using a predetermined statistical plan. The full analysis set (FAS) included patients receiving at least one dose of durvalumab, one dose of cisplatin, and one session of CIRT. The population of patients that complied with the guidelines for GCP was defined as the safety analysis set (SAS), and AEs in the SAS were evaluated [39]. The per-protocol set (PPS) was the population of patients excluded from the FAS who severely violated the study protocol provisions. The PPS was used to assess treatment effectiveness. Prior to data analysis, the study personnel determined eligibility for the FAS, SAS, and PPS.

All AEs were evaluated using the Common Terminology Criteria for Adverse Events version 5.0 [40]. The DLTs were assessed from the start of treatment until 92 days later. An independent data-monitoring committee confirmed all AEs and DLTs.

Regarding efficacy analyses, the OR was the percentage of patients with CR or partial response among the PPS that could be evaluated, as confirmed based on Response Evaluation Criteria in Solid Tumors version 1.1 [41]. The evaluation was based on imaging assessments, including CT and MRI. Figure 2 includes the time points for each imaging diagnosis. CR means the disappearance of all target and non-target lesions as determined at the 28-week assessment in this study. No interim analysis was performed for safety or efficacy in the present study.

## 5. Conclusions

In conclusion, we reported the early safety and efficacy results of combining chemo-CIRT and durvalumab for locally advanced cervical cancer. The long-term follow-up of this dataset and the immune response in these patient samples will be reported in future articles.

## Figures and Tables

**Figure 1 ijms-24-10565-f001:**
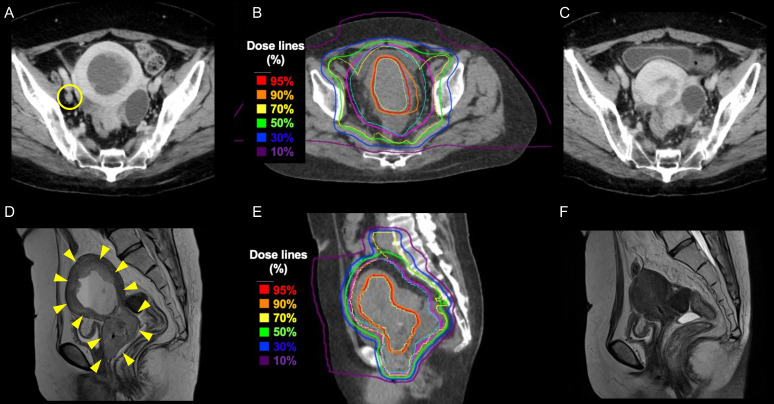
Representative images of the patients who received the treatment. (**A**) Pre-treatment axial computed tomography (CT) image. The yellow circles indicate enlarged metastatic lymph nodes. (**B**) Dose distribution on axial CT. Enlarged metastatic lymph nodes were not observed. (**C**) Axial CT image obtained three months after treatment. (**D**) Pre-treatment sagittal magnetic resonance imaging (MRI), T2 weighted image (WI). The tumor invades the area circled by the yellow arrows. (**E**) Dose distribution on sagittal CT. (**F**) Sagittal MRI, T2 WI three months after treatment. There was no evidence of residual tumor.

**Figure 2 ijms-24-10565-f002:**
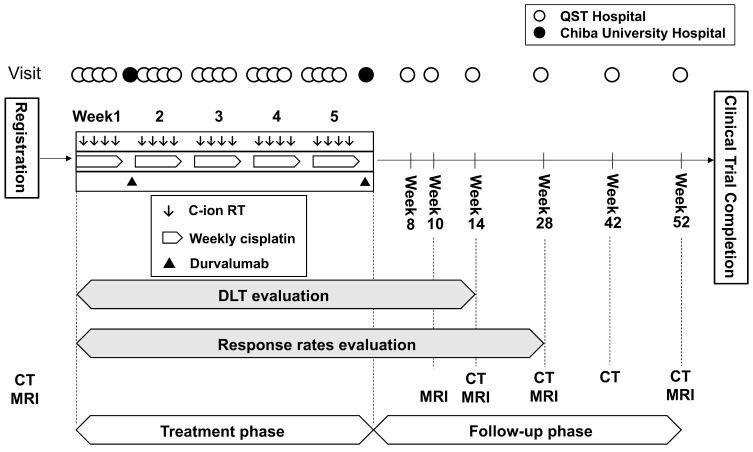
Design of the present study. C-ion RT = carbon-ion radiotherapy; DLT = dose-limiting toxicity; CT = computed tomography; MRI = magnetic resonance imaging.

**Table 1 ijms-24-10565-t001:** The clinical characteristics of the three patients.

	Case 1	Case 2	Case 3
Age at the enrollment	7X	4X	5X
ECOG PS	0	0	0
Race and ethnicity	Japanese	Japanese	Japanese
Staging, FIGO (2018)	IIIC1r	IIIC1r	IIIC1r
Staging, TNM UICC (8th)	cT2bN1M0	cT2bN1M0	cT3bN1M0
Tumor histology	Squamous cell carcinoma, NOS	Endocervical adenocarcinoma, usual type	Adenocarcinoma, NOS
Maximal tumor diameter, mm	60	46	107
Past medical history	Right wrist fracture	Asthma, cerebral aneurysm, and psoriasis	(None)
Coexisting disease	Headache	Menopausal disorders	Depression, conjunctival hyperemia, hyperlipidemia, and hay fever

ECOG PS = Eastern Cooperative Oncology Group performance status; FIGO = the International Federation of Gynecology and Obstetrics; TNM = Tumor, node, and metastasis; UICC = Union for International Cancer Control; NOS = not otherwise specified.

**Table 2 ijms-24-10565-t002:** Adverse events (possibly, probably, or definitely) attributed to the treatment.

Items		Grading, Number of Patients			
	Grade 0	Grade 1	Grade 2	Grade 3	Grade ≥ 4
Hematologic toxicity					
Leukopenia	0	0	1	2	0
neutropenia	0	0	0	3	0
Non-hematologic toxicity					
Hypothyroidism	2	0	0	1	0
Stomatitis	2	1	0	0	0
Colitis/Diarrhea	3	0	0	0	0
Cystitis	3	0	0	0	0
Dry mouth and eyes *	2	1	0	0	0
Eczema	1	1	1	0	0
Creatinine increased	2	0	1	0	0
Serum amylase increased	2	0	1	0	0
GGT increased	1	0	1	1	0

GGT = gamma-glutamyl transpeptidase. * This patient was diagnosed with Sjogren’s syndrome.

**Table 3 ijms-24-10565-t003:** The dose-limiting toxicities of the present study.

Hematologic Toxicity
- Grade ≥ 3 neutropenia complicated by fever ≥ 38.3 °C - Grade 4 neutropenia (≥7 days) - Grade ≥ 3 thrombocytopenia with significant bleeding - Grade 4 thrombocytopenia (regardless of duration) - Grade 4 anemia (regardless of duration)
Non-Hematologic Toxicity
- Any Grade 4 non-immune-mediated AE - Any Grade 4 immune-mediated AE, excluding endocrinopathies - Any Grade 3 non-immune mediated AE that does not resolve to ≤Grade 1 or baseline within 30 days with optimal medical management - Any Grade 3 immune-mediated AE—excluding diarrhea/colitis, pneumonitis, hepatitis, rash, neurotoxicity, myocarditis, myositis/polymyositis, endocrinopathies, and nephritis—that does not resolve to ≤Grade 1 or baseline within 30 days after onset of the event despite optimal medical management including systemic corticosteroids - Grade 3 diarrhea or colitis that does not resolve to ≤Grade 1 within 14 days [both immune- and non-immune-mediated indicated here; the same is the case if not specified in the remaining bullet points below] - Grade 3 non-infectious pneumonitis - Grade 2 non-infectious pneumonitis that does not resolve to ≤Grade 1 within 3 days of the initiation of maximal supportive care - AST or ALT ≥ 3 × ULN with a concurrent increase in TBL ≥ 2 × ULN without evidence of cholestasis or alternative explanations (e.g., viral hepatitis, disease progression in the liver, i.e., “Hy’s Law”) - ALT or AST > 8 × ULN or TBL > 5 × ULN - Grade 3 immune-mediated rash that does not resolve to ≤Grade 1 or baseline within 30 days - Grade 2 rash covering > 30% BSA that does not resolve to ≤Grade 1 or baseline within 30 days - Any grade of immune-mediated rash with bullous formation - Grade 3 immune-mediated neurotoxicity (excluding Guillain-Barre and myasthenia gravis) that does not resolve to ≤Grade 1 within 30 days - Grade 2 or 3 immune-mediated peripheral neuromotor syndrome (such as Guillain-Barre and myasthenia gravis) that does not resolve to ≤Grade 1 within 30 days or that exhibits signs of respiratory insufficiency or autonomic instability - Grade 3 immune-mediated myocarditis - Any symptomatic immune-mediated myocarditis that does not become asymptomatic within 3 days of initiating optimal medical management, including systemic corticosteroids - Grade 2 or 3 immune-mediated myositis/polymyositis that does not resolve to Grade ≤ 1 within 30 days of initiating optimal medical management, including systemic corticosteroids, or that exhibits signs of respiratory insufficiency regardless of optimal medical management - Immune-mediated increase in creatinine >3 × ULN, or >3 × baseline for patients with a baseline creatinine elevated above ULN

AE = adverse event; AST = aspartate aminotransferase; ALT = alanine aminotransferase; UNL = upper limit of normal; TBL = total bilirubin; BSA = Body surface area.

## Data Availability

The data is unavailable because the data related to this study belong to AstraZeneca.

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
