# Peer review of "A Phase Ib Study of Durvalumab (MEDI4736) in Combination with Carbon-Ion Radiotherapy and Weekly Cisplatin for Patients with Locally Advanced Cervical Cancer (DECISION Study): The Early Safety and Efficacy Results"

_ijms, 2023, doi:10.3390/ijms241310565_

Round 1
Reviewer 1 Report
The authors show the results of a new combination of therapies in three patients. This is a small number of patients, as the authors also recognize, but the provided results are straightforward and clearly presented. The only minor comment would be to add to the conclusion of the abstract that further research is required as only three patients were included in this study.
The English language is fine. At some places an article (e.g. a or the) might be added.
Author Response
We wish to thank the reviewer for reviewing our paper and providing positive feedback. Please see the attached PDF.

Reviewer 2 Report
I am grateful for the opportunity to review manuscript ID: ijms-2447321, entitled “A phase Ib study of durvalumab (MEDI4736) in combination with carbon-ion radiotherapy and weekly cisplatin for patients with locally advanced cervical cancer (DECISION study): the early safety and efficacy results”.
The authors present their findings on chemo-immuno-particle-therapy of locally advanced cervical cancer in the context of a phase Ib clinical trial. The work fit the scope of the journal. The manuscript is clear, the methods are appropriate, the conclusions are consistent with the presented results.
A clear strength of the work is its novelty in combining immune checkpoint inhibition with carbon ion therapy.
Although the trial could not be completed per protocol, it is worth to publish the results of the DLT-evaluation cohort of three cases. It would be fair to declare the very low patient number (practically a case series) as a major limitation of this study in the discussion part of the article.
Specific comments:
In general, please consider using conventional article structure (introduction-methods-results-discussion).
Please consider elaborating which factors of COVID-19 made the enrollment difficult. Were these linked to the accessibility of CIRT, or any logistic issues? Interpolating to post-COVID era, could any of these factors influence the feasibility of the chemo-immuno-particle-therapy? (As your aim is to report feasibility, I suppose.)
You report 100% grade 3 neutropenia. Influenced this toxicity the chemotherapy regimen?
Immune-mediated and hematologic early toxicity are reported, but radiation toxicity is missing. Please consider reporting late (1-year, per protocol) toxicity, giving the readers a complete overview of the primary endpoints. Reporting late toxicities for a case series of three as a separate work would be reduced firm and sound.
Please elaborate which method was used to determine complete response. Assuming pathologic response was not examined, maybe radiologic response was used. Has CR been defined according to RECIST 1.1? Which diagnostic radiology modality (modalities) in which time point has (had) been used?
I’d suggest replacing citation 41 (RECIST 1.1): Eisenhauer EA, Therasse P, Bogaerts J, Schwartz LH, Sargent D, Ford R, Dancey J, Arbuck S, Gwyther S, Mooney M, Rubinstein L, Shankar L, Dodd L, Kaplan R, Lacombe D, Verweij J. New response evaluation criteria in solid tumours: revised RECIST guideline (version 1.1). Eur J Cancer. 2009 Jan;45(2):228-47. doi: 10.1016/j.ejca.2008.10.026. PMID: 19097774.
Author Response
We wish to thank the reviewer for reviewing our paper and providing careful and constructive feedback. Please see the attached PDF.

Round 2
Reviewer 2 Report
Thank you for revising the article.